

# Overexpression of a *Rosa rugosa* Thunb. *NUDX* gene enhances biosynthesis of scent volatiles in petunia

Lixia Sheng, Shu Zang, Jianwen Wang, Tiantian Wei, Yong Xu and Liguo Feng

College of Horticulture and Plant Protection, Yanghzou University, Yangzhou, Jiangsu, China

## ABSTRACT

*Rosa rugosa* is an important natural perfume plant in China. Rose essential oil is known as 'liquid gold' and has high economic and health values. Monoterpenes are the main fragrant components of *R. rugosa* flower and essential oil. In this study, a member of the hydrolase gene family *RrNUDX1* was cloned from Chinese traditional *R. rugosa* 'Tang Hong'. Combined analysis of *RrNUDX1* gene expression and the aroma components in different development stages and different parts of flower organ, we found that the main aroma component content was consistent with the gene expression pattern. The *RrNUDX1* overexpressed *Petunia hybrida* was acquired via *Agrobacterium*-mediated genetic transformation systems. The blades of the transgenic petunias became wider and its growth vigor became strong with stronger fragrance. Gas chromatography with mass spectrometry analysis showed that the contents of the main aroma components of the transgenic petunias including methyl benzoate significantly increased. These findings indicate that the *RrNUDX1* gene plays a role in enhancing the fragrance of petunia flowers, and they could lay an important foundation for the homeotic transformation of *RrNUDX1* in *R. rugosa* for cultivating new *R. rugosa* varieties of high-yield and -quality essential oil.

## INTRODUCTION

*Rosa rugosa* Thunb., which usually serves as a good plant material for landscaping, is one of the oldest natural perfume plants. The rose essential oil extracted from its flowers is expensive and is mainly used in the high-end perfume, cosmetic, and health care industries (*Ma et al., 2004*). The demand for rose essential oil and its high fragrance quality is increasing in European and Asian markets. Besides optimization of extraction equipment and techniques, breeding *R. rugosa* variety rich in high-yield and -quality essential oil is the preferred way to overcome the high demand of rose essential oil.

Improving rose essential oil attributes by biotechnological breeding requires a well understanding of the biosynthesis of main aroma components in essential oil. Terpenes are the most abundant volatiles in floral aroma, including monoterpenes, sesquiterpenes and diterpenes, etc. (*Pichersky & Dudareva, 2007*). In all aroma components of *R. rugosa* flowers, monoterpenes decide the content and quality of rose essential oil to a great extent. Monoterpenes such as citronellol, geraniol, nerol and their derivatives of acetate esters

Corresponding authors
Yong Xu, Yongxu@yzu.edu.cn
Liguo Feng, lgfeng@yzu.edu.cn

that account for 50%–70% of the overall mass fraction of rose essential oil are the main aroma components of rose essential oil (*Feng et al., 2010*; *Magnard et al., 2015*). Therefore, increasing monoterpenes synthesis in *R. rugosa* flowers would be an effective way to increase the content of aroma components in rose essential oil. The precursor of terpene isoprene pyrophosphate (IPP) was synthesized via the mevalonate (MVA) pathway in the cytoplasm and methylerythritol 4-phosphate (MEP) pathway in the plastid (*Bick & Lange, 2003*; *Sapir-Mir et al., 2008*; *Simkin et al., 2011*; *Lange & Ahkami, 2013*). Interestingly, the enzymatic machinery of geraniol biosynthesis in modern rose *(Rosa × hybrida)* is different with the classic pathway induced by a terpene synthase (*Magnard et al., 2015*). The gene *Nucleoside diphosphates linked to some X moiety* (*NUDX1*) play a key role in the fragrance formation revealed a unique pathway: The NUDX cuts down a phosphate group of substrate geranyl diphosphate (GPP) to produce geranyl monophosphate (GP) and then geraniol and other monoterpenes form from dephosphorylation of GP by an unknown phosphatase (Fig. S1) (*Tholl & Gershenzon, 2015*).

The *NUDX* gene belongs to the Nudix hydrolase family and is widely found in various organisms, including bacteria, yeasts, algae, nematodes, vertebrates and plants (*Bessman, Frick & O'Handley, 1996*; *Xu et al., 2004*; *Kraszewska, 2008*). These enzyme families share a conserved Nudix motif $GX_5EX_7REVXEEXGU$ (*Ogawa et al., 2008a*). Nudix hydrolase performs a role in regulation and signal transduction in plant stress (*Bessman, Frick & O'Handley, 1996*; *Xu et al., 2004*). It repairs the oxidative damage of guanine metabolism on DNA and protects the nucleic acid metabolic reactions in plants (*Ogawa et al., 2009*; *Yoshimura & Shigeoka, 2015*). Recent studies on the Nudix hydrolase gene is mainly focused on the model plant *Arabidopsis thaliana* (L.) Heynh., which possesses a total of 29 Nudix hydrolase genes (*Yoshimura & Shigeoka, 2015*). On the basis of predicted subcellular localization, the encoded proteins can be divided into three types: *AtNUDX1 to −11*, *AtNUDX12 to −18*, or *AtNUDX19 to −24* (*Ogawa et al., 2005*; *Yoshimura et al., 2007*). In the *AtNUDX* gene family of Arabidopsis, overexpression of *AtNUDX1* can attenuate or repair DNA and RNA oxidative damage (*Yoshimura et al., 2007*; *Ogawa et al., 2008b*). Overexpression of *AtNUDX2* gene could increase oxidative stress tolerance in Arabidopsis (*Ogawa et al., 2009*). As a positive regulatory protein in the signal transduction pathway of salicylic acid on which non-expressor of pathogenesis-related genes1 (NPR1) is dependent, *AtNUDX6* significantly affects the plant immune response (*Ishikawa et al., 2010a*; *Ishikawa et al., 2010b*). Meanwhile, the *AtNUDX7* gene negatively regulates this pathway (*Ishikawa et al., 2010a*; *Ishikawa et al., 2010b*).

In the present study, Chinese traditional *R. rugosa* was taken as a test material for cloning, subcellular localization, temporal and spatial expression analysis of *RrNUDX1* gene which is the ortholog of *RhNUDX1* (Figs. S1 –S2, (*Magnard et al., 2015*). By correlation analysis of main volatile monoterpenes in developmental stages and different parts of flower organs and transgene functional analysis, we prefer to investigate the function associated with volatile monoterpenes of *RrNUDX1* gene. The results could lay the foundation for understanding the mechanism of floral fragrance regulation and provide gene resources for creating rose germplasm with high essential oil.

## MATERIALS & METHODS

### Plant materials

*Rosa rugosa* 'Tanghong', a representative Chinese traditional rose cultivar, was used as experimental material. The three-year-old clonal seedings planted in the field of Rose Resources of Yangzhou University (N32°23′27.64″, E119°25′10.23″) were selected as the flower source. According to previous division standard, the petals of flowers during five stages and different parts of full opening flowers were picked (*Feng et al., 2014*). For every sample, 3 biological replications were prepared with different ramets and 3–5 flowers of the same plant were selected for each replication.

### RNA extraction and purification

Total RNA was isolated from different *R. rugosa* tissues according to the manufacturer's instructions of MiniBEST Universal RNA Extraction Kit (TaKaRa, Japan). RNA samples were treated with DNase using DNaseI kit (TaKaRa, Japan) according to the manufacturer's guidelines, and then quantified by a spectrophotometer (Eppendorf, Germany) at 230 nm, 260 nm and 280 nm.

### Cloning of *RrNUDX1* gene and sequence analysis

Full cDNA of *RrNUDX1* gene was cloned by rapid amplification of cDNA end (RACE). The protocols of 3′ RACE and 5′ RACE were same as previously described in *Feng et al. (2014)* with the 3′ RACE nest primers (5′-AGCCAAACCATCGCAGTA-3′ for first round 5′-ATGGTTGGGGATGGTATG-3′for second round) and 5′ RACE gene-specific primer (5′-CTGCTGTGCCAATGCTGA-3′). The full-length cDNA of the *RrNUDX1* gene was assembled and analyzed using DNAMAN 5.0 software.

### Subcellular localization of *RrNUDX1* gene

The cDNA was synthesized from 1 μg RNA using PrimeScript RT reagent Kit with gDNA Eraser (TaKaRa, Japan). Complete CDS of *RrNUDX1* were isolated from the cDNA. CDS of *RrNUDX1* and pBWA(V)HS-GFP vectors were digested by BsaI/Eco31I enzymes, and pBWA(V)HS-*NUDX1*-GFP fusion expression vector was constructed by using $T_4$DNA ligase connection, empty vector pBWA(V)HS-GFP was used as control. The two vectors were transferred into the competent cells of *Agrobacterium tumefaciens* LBA4404 by electroporation in MicroPulser electroporator (Bio-RAD, USA) at voltage of 2.4 KV/5 ms, and the positive bacteria were screened by Kanamycin and PCR. Then the screened positive bacteria were expanded in LB liquid medium to $OD_{600} = 0.6$, and the lower epidermis of tobacco leaves was injected and infected with these bacteria fluid, the tobacco plants were cultured in low light. After 2 days, tobacco leaves were taken and the fluorescence of the transformed leaf cells was imaged using a confocal laser-scanning microscope (Olympus FV10 ASW).

### Temporal and spatial expression analysis

The relative expression of *RrNUDX1* were analyzed by real-time quantitative reverse transcription PCR (qRT-PCR). *Rosa hybrid* $\alpha$- tubulin subunit actin gene

(GenBank accession no. AF394915.1) was used as an internal reference gene (Actin-F: 5′-GCCACCATCAAGACCAAG-3′; Actin-R: 5′-ATCAATGCGGGAGAACAC-3′). Experiments of qPCR were performed with 12.5 µl SYBR® *Premix Ex Taq* (2×) (TaKaRa, Japan), 1 µl Forward Primer (10 µM, 5′-GCGGTGGTAGTATGCCTGTT-3′), 1 µl 10 µM Reverse Primer (10 µM, 5′-TTCCTTCAGTTCCCTTGCTG-3′), 2 µl cDNA and 8.5 µl ddH$_2$O. PCR procedure is an initial incubation at 95 °C for 5 min, then followed by 40 cycles of 15 s at 95 °C, 30 s at 60 °C, and 30 s at 72 °C on a BIO-RAD CFX96 Real-Time System (Bio-Rad, USA). The expression level calculation according to the comparative threshold cycle (Ct) (2- *Delta Delta*Ct) method (*Schmittgen & Livak, 2008*) followed previous description (*Feng et al., 2014*).

## Gas chromatography with mass spectrometry (GC-MS) analysis of headspace volatiles

1 g of fresh flower tissue and internal standard (3-nonanone, 0.8 µg µL$^{-1}$, Sigma, USA) were used for the headspace solide-phase microextraction for each sample. The extraction protocol and GC-MS protocol were the same as our previous method with minor parameter adjustments (*Feng et al., 2010*; *Feng et al., 2014*). The adjustments included mass spectra scanning at m/z 30–600 amu, more sensitive FFAP elastic quartz capillary vessel column (60 m ×0.32 mm I.D., 1.0µm film, Agilent Corporation, USA) and column temperature program (initial temperature at 50 °C for 1 min, and then increased at 5 °C/min to 120 °C, then increased at 8 °C/min to 200 °C, finally increased at 12 °C/min to 250 °C which was maintained for 7 min).

## Qualitative and quantitative analysis of headspace volatiles

Significant data selection by the Xcalibur (a shareware of Thermo Electron Corporation, https://xcalibur.updatestar.com/) and quantitative analysis of the headspace compounds with internal standard 3-nonanone (0.8 µg µL$^{-1}$) were performed as lab self-built method (*Feng et al., 2010*).

## Overexpression of *RrNUDX1* gene in Petunia hybrida

The recombinant pCAMBIA1304-*RrNUDX1* expression vector construction and transfection into *A. tumefaciens* EHA105 competent cell (TransGen, China) followed previous method (*Sheng et al., 2018*). Transformation to *Petunia hybrida* cv 'Mitchell Diploid' via *Agrobacterium*-mediated infection on leaf disc were refers to *Guo et al. (2014)*. Petunia explant leaf discs (0.5 cm diameter) cut from 4-week-old seedlings of aseptic seed were pre-cultured on pre-cultured MS medium (30 g/L sucrose, 7 g/L Agar, 3.0 mg/L 6-BA and 0.2 mg/L IAA) at 28 °C for 2 d, then co-incubated with infection liquid where positive EHA105 (OD$_{600}$ 0.8–1.0) was resuspended to OD$_{600}$ = 0.3. After co-incubation for 5 min, the explants with no residual infection liquid were cultured on pre-cultured medium with additional 30 µMol/L acetosyringone in dark for 2 days. Explants were transferred to the screening medium (pre-cultured medium with suited selection pressure) to induce callus and the medium was renewed every 2 weeks to subculture resistant calluses. When the seedlings (2–3 cm adventitious buds) regenerated from the callus, the buds were cut off to subculture on the rooting 1/2 MS medium (30 g/L sucrose, 7 g/L agar, 0.1 mg/L NAA) with

selection pressure. The hygromycin selection pressures were 7 mg/L and 6 mg/L at callus induction and buds rooting, respectively, and bacteriostat was 500 mg/L carbenicillin. The regeneration seedings growing up to five leaves with one sprout were transplanted into soil. The culture condition was 16 h light period, light intensity of 200 $\mu$mol m$^{-2}$s$^{-1}$, temperature of 25 °C/23 °C and relative humidity of 70%. Callus GUS staining and PCR were used to select the genomic transgenic petunia. The overexpression level of *RrNUDX1* in transgenic lines were detected by qRT-PCR. GC–MS was used to analyze the aroma composition and content of wild–type and transgenic petunia flowers in bloom.

## Mathematics statistical methods

The average value of three replicates with standard error (SE) was used as data of each sample. In significance difference test between two sample, independent two-sample *t*-test was performed for two data groups. In significance difference test of pairwise comparison of more than three samples, one-way ANOVA (Analysis of Variance) and LSD (Least Significant Difference) was performed for multiple comparison. All the calculation was based on the SPSS 18.0 (IBM SPSS Modeler 18.0, https://www.ibm.com/support/pages/downloading-ibm-spss-modeler-180).

# RESULTS

## Isolation, sequence analysis and subcellular localization of RrNUDX1

We successfully isolated a NUDX1 gene (*RrNUDX1*, GenBank accession number KX096710.1) related to the monoterpenes biosynthesis of R. rugosa. The full cDNA of *RrNUDX1* was 777 bp in length including a 453 bp coding sequence (CDS) which encodes 151 amino acids, the 5′ untranslated region (UTR)(68 bp) and the 3′ UTR (224 bp). *RrNUDX1* had high (98%) sequence identity with homology gene of *Rosa ×hybrida* 'Papa Meilland' (*RhNUDX1*, JQ820249.1) with only 7 SNPs including 3 nonsynonymous SNPs in the CDS. RrNUDX1 protein had higher identity with homology protein of *Rosa chinensis* (AFW17224.1) (98.67%) than RhNUDX1 (M4I1C6.1) (98%) (Fig. S2). RrNUDX1-GFP fusion protein exhibited no signal in the nucleus of tobacco leaf cells, indicating that *RrNUDX1* localized in the cytoplasm (Fig. 1).

## *RrNUDX1* expression and volatile monoterpenes accumulation in flower

Temporal and spatial expression of *RrNUDX1* had a significant difference from budding to senescence in *R. rugosa* 'Tanghong' (Figs. 2A–2D). The expression level of *RrNUDX1*, which was low in the budding phase, rapidly increased from the early opening stage, reached the highest level at the half-opening stage, started to decrease from the blooming stage to 49.806% of that during the half-opening stage, and then markedly decreased during the senescence phase. The expression level of *RrNUDX1* exhibited tissue specificity from the different organs of the flower, and the expression level reached the highest in the petals. However, the expression levels in the stamen and pistil were only 6.9% and 5.2% of that in the petals, respectively. Meanwhile, the expression levels in the anthocaulus, receptacle, and calyx were extremely low.
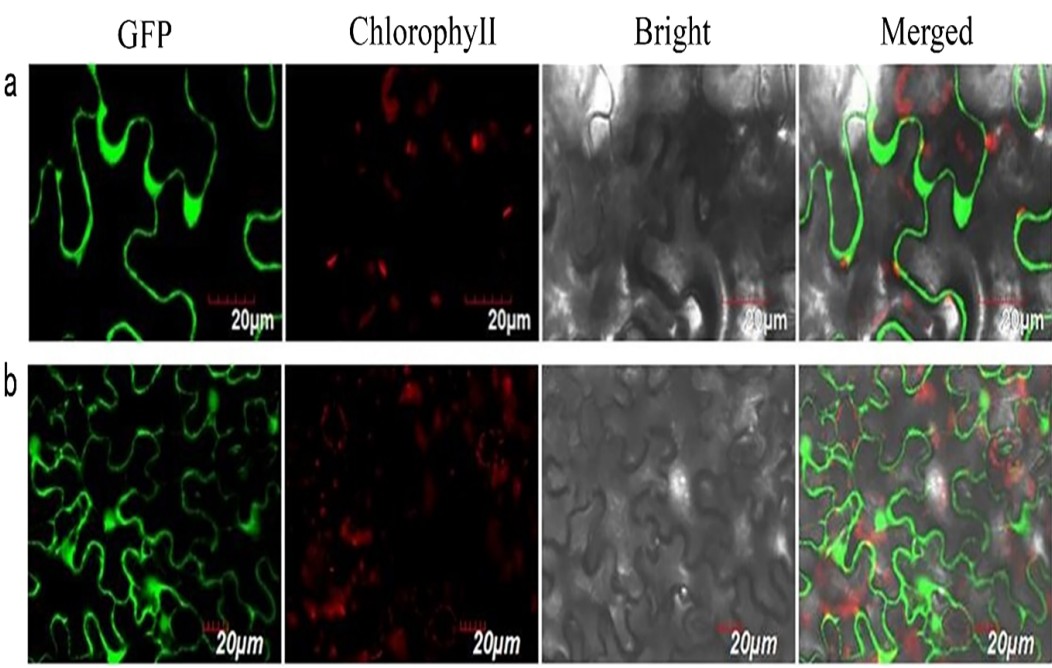

**Figure 1** **Subcellular localization of *RrNUDX1*.** Fluorescence signals were visualized using confocal laser-scanning microscopy. Green fluorescence indicates GFP, red fluorescence indicates chloroplast aut-ofluorescence (A) 35S: RrNUDX1-GFP; (B) 35S: GFP.

To verify the relations between the gene expression of *RrNUDX1* and the accumulation of the main monoterpenoids in *R. rugosa*, we measured and determined the aroma component and content. Emphatical analysis was conducted on representative components, such as geraniol, citronellol, nerol (Figs. 2E–2F), and their acetate derivatives, including geraniol acetate, citronellyl acetate, and neryl acetate (Figs. 2G–2H). The contents of the main monoterpenes and their acetate derivatives initially increased and then decreased with the flower development. Geraniol, nerol, geranyl acetate, and neryl acetate reached the highest levels in the blooming stage, whereas citronellol and citronellyl acetate reached their highest levels during the half-opening stage. The total content of the six main aromatic components in the half-opening stage was 10.12 μg/g higher than that in the blooming stage. The six main aromatic components mainly originated from the petal and stamen. The amount of the three monoterpene alcohols in the petal was more than four (4.34) times that in the stamen, whereas the amount of the three acetate derivatives in the stamen was 192.21 μg/g, which was more than three times that in the petal. Several aroma components were detected in the stamen, whereas no monoterpene alcohols or acetate derivatives mentioned above were found in the anthocaulus, receptacle, and calyx.

## Overexpression vector construction of RrNUDX1 gene and genetic transformation to petunia hybrida

The petiole callus transformed by 35S: *RrNUDX1* vector were stained with varying degrees of blue by GUS staining (Fig. S3). *RrNUDX1* was successfully integrated into the genome

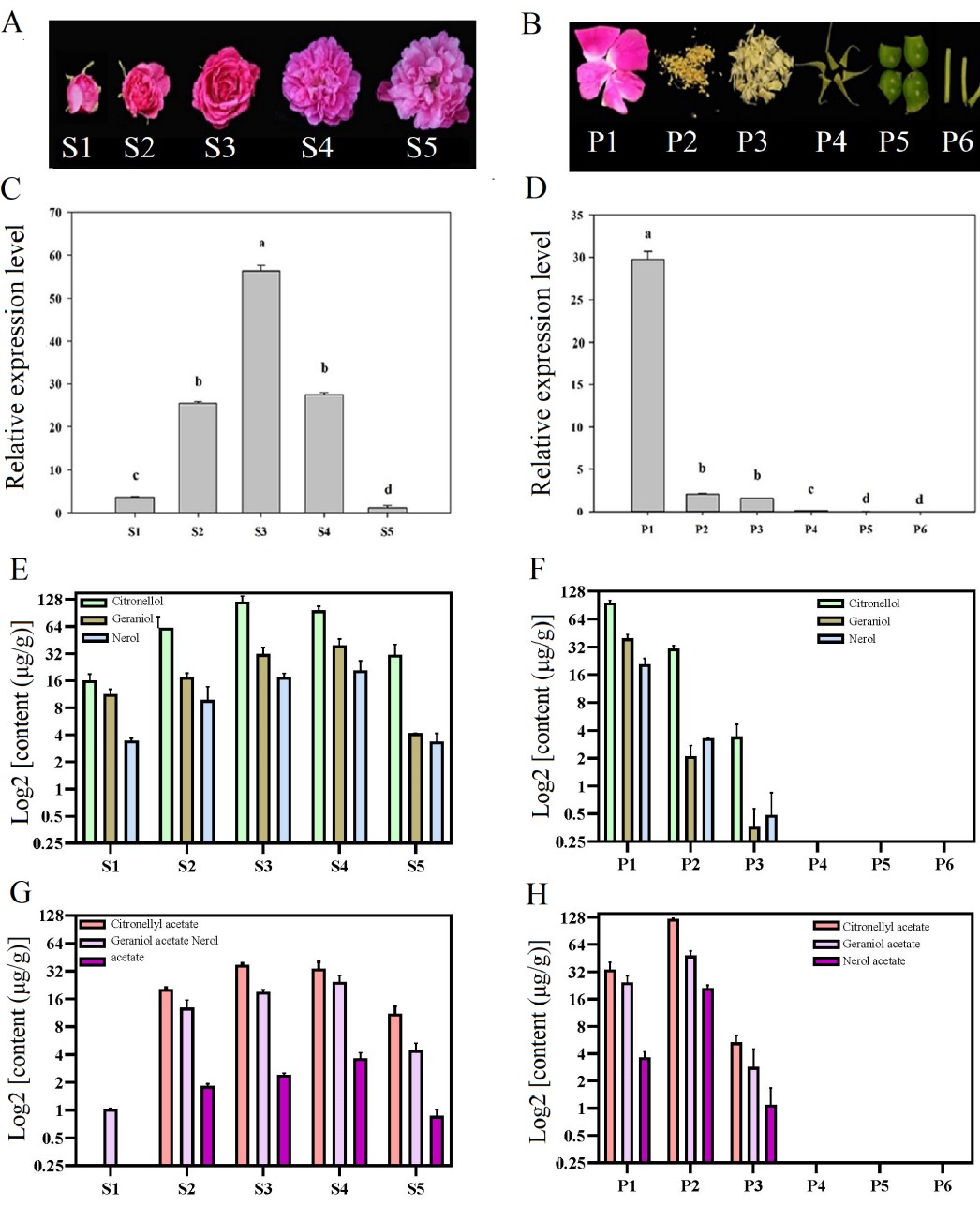

**Figure 2** **Expression of *RrNUDX1*. and major volatile components in different development stages (A) and different parts of flower organ (B) in *R. rugosa* 'Tanghong'. (C) and (D) Relative expression level of *RrNUDX1*; (E)–(H) Major volatile components.** S1: budding stage, S2: early opening stage, S3: half opening stage, S4: full opening stage, S5: withering stage; P1: petal, P2: stamen, P3: pistil, P4: calyx, P5: receptacle, P6: pedicle. Values represent the means ± SE. Lowercase letters (a, b, c) stand for significantly different (LSD test, *P* < 0.05).

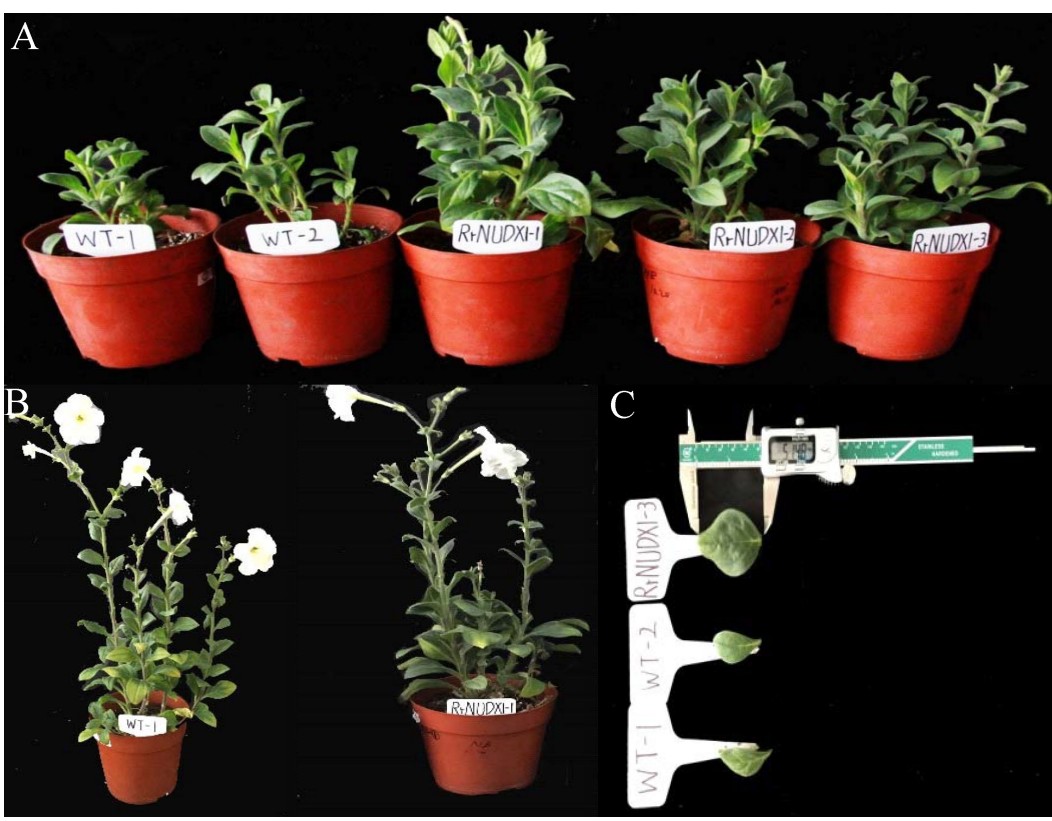

**Figure 3** **Phenotype of transgenic *petunia hybrida*.** WT1-2: Wild type; *RrNUDX1* 1-3: Transgenic petunia Line1, line2 and Line3. (A) Phenotype of wild and transgenic petunia plants after 35 days of transplanting; (B) A wild type petunia under flowering and a transgenic petunia under flowering; (C) Leaf morphology of wild and transgenic petunia plants after 35 days of transplanting.

**Table 1** **Comparison of leaf morphology between wild and transgenic plantsafter 35 days of transplanting.**

| Index | Wild type (mm) | Transgenetic plants (mm) |
|---|---|---|
| Leaf length | 36.42 ± 0.99b | 47.25 ± 4.49a |
| Leaf width | 17.39 ± 1.34b | 25.58 ± 6.45a |
| Aspect ratio | 2.1 ± 0.11a | 1.84 ± 0.13b |

**Notes.**
Values represent the means ± SE. Different letters (a, b, c) stand for significantly different (LSD test, $P < 0.05$).

of five petunia lines by PCR detection of six randomly selected lines (Fig. S4) and expressed significantly (Fig. S5). The wild-type and transgenetic petunia plants significantly differed in phenotype under the same conditions. Compared with those of the wild type, the leaves of the petunia with *RrNUDX1* gene transformed were significantly wider, with 10.83 mm and 8.19 mm larger width and length than the wild type; moreover, the transgenic plants grows faster compared with the wild-type plants (Fig. 3, Table 1).

**Table 2  Comparative analysis of floral components in wild type and transgenic petunia.**

| Components | Content (ug/g) | | | |
|---|---|---|---|---|
| | *WT* | *Line1* | *Line2* | *Line3* |
| Tetradecane | 0.08 ± 0.03c | 0.18 ± 0.07b | 0.18 ± 0.01b | 0.25 ± 0.03a |
| Methyl benzoate | 37.32 ± 0.86b | 63.04 ± 2.22a | 52.24 ± 11.66a | 39.53 ± 1.02b |
| Phenylmethyl acetate | 0.13 ± 0.02b | 0.20 ± 0.02a | 0.21 ± 0.03a | 0.11 ± 0.03b |
| Methyl salicylate | 0.21 ± 0.03b | 0.35 ± 0.07a | 0.31 ± 0.03a | 0.30 ± 0.07a |
| Benzyl butyrate | 0.24 ± 0.01b | 0.67 ± 0.18a | 0.64 ± 0.39a | 0.25 ± 0.00b |
| Benzyl benzoate | 5.94 ± 1.91ab | 8.91 ± 1.22a | 5.23 ± 3.42b | 3.79 ± 0.08b |
| Benzyl alcohol | 0.43 ± 0.21a | 0.54 ± 0.13a | 0.66 ± 0.38a | 0.33 ± 0.14a |
| Phenethyl alcohol | 0.41 ± 0.01a | 0.57 ± 0.03a | 0.43 ± 0.29a | 0.28 ± 0.18a |
| Eugenol | 0.35 ± 0.06b | 0.98 ± 0.34a | 0.50 ± 0.31b | 0.44 ± 0.03b |
| Soeugenol | 0.10 ± 0.04c | 0.32 ± 0.06b | 0.43 ± 0.08a | 0.49 ± 0.10a |

**Notes.**

Values represent the means ± SE. Different letters (a, b, c) stand for significantly different (LSD test, $P < 0.05$).

GC–MS were used to detect the aroma components of the petunia flowers in bloom, and 10 major aroma components were selected for statistical analysis (the sum of the 10 aroma components accounted for 80.10% of the total aroma components in petunia) (Table 2). Results showed that the contents of all ester fragrance ingredients, except for benzyl benzoate, increased in the transgenic petunia plants. In particular, the increase in the content of methyl benzoate was the most significantly. In the transgenic Line1, the content of methyl benzoate was 63.04 µg/g, which was 1.69 times that in the wild-type plants. The alcohols were mainly benzyl alcohol and phenylethyl alcohol, and the contents of benzyl alcohol in transgenic Line1 and Line2 were 25.58% and 53.49% higher than those in the wild-type plants. In addition, the contents of the two main phenolic compounds also increased, and the contents of isoeugenol in the three transgenic Lines were 3.2, 4.3, and 4.9 times those in the wild-type plants.

## DISCUSSION

Floral substances are secondary metabolites released from plant flowers, these substances are mainly composed of numerous low molecular-weight volatile compounds (*Pichersky, Noel & Dudareva, 2006*) that can attract insect pollination, improve the aesthetic value of ornamental plants, and enhance the quality and economic value of flower products (*Negre-Zakharov, Long & Dudareva, 2009*). Geraniol is the main aroma component of *Rosa rugosa* flower and essential oil, it is formed in plants through two pathways. One is the MEP pathway in plastids, where upstream GPP was utilized by GES to form geraniol through catalysis and then forms nerol under isomerase action (*Feng et al., 2014*). The expression level of GES directly affects the yield of geraniol and consequently exerts a controlling effect on the synthesis of downstream indole monoterpenoid alkaloids (*Kumar et al., 2015*). Another special way to form geraniol is with the participation of *NUDX1*. *NUDX1* and *GES* have similar functions in geraniol synthesis, both of them have the same precursor substance GPP, and can express in the cytoplasm for producing geraniol

glycosides and geraniol (*Magnard et al., 2015*). In this study, *RrNUDX1* gene related to the biosynthesis of main aromatic components of Chinese traditional *R. rugosa* was investigated. Generally, the total aroma components during the bloom stage was consistent with *RrNUDX1* expression, which initially increased and then decreased. Interestingly, the citronellol (and citronellyl acetate) reached the highest level in the half-opening stage unlike other components in the full opening stage and were more coincident with the *RrNUDX1* mRNA accumulation than geraniol. The reason of unsynchronized peak time may be complex and we conjectured it should involve in other genes of monoterpene synthesis or conversion. On the one hand, the increasing geraniol (and its isomeride nerol) provide substrates of dehydrogenation for citronellol accumulation (*Feng et al., 2014*). One the other hand, potential GES of citronellol may decide on the citronellol accumulation on to a greater extent by inducing citronellol synthesis. Whereas, the studies of dehydrogenases of geraniol or nerol are greatly about geraniol acid and haven't identified the one for citronellol transformation until now (*Hassan et al., 2012*; *Tan et al., 2019a*; *Tan et al., 2019b*), much less to the indistinguishable identification of geraniol or citronellol GES which is hidden in ambiguous function of terpenoid synthase family (*Magnard et al., 2015*). We prefer that the high citronellol accumulation more result from the induced response of an assumed GES. Though lack of strong evidence for our conjecture, our ongoing study found that several *TPSs* (*GES* candidates) with similar expression pattern with *RrNUDX1* except in stamen and half opening stage while whether the induction would happen by these candidate GES is on testing (Unpublished). In addition, the aroma components of *R. rugosa* mainly concentrated in the petals, stamens, and pistils, and their contents were extremely low in the other parts, this is consistent with the expression of *RrNUDX1* gene in different parts of floral organ. These results indicated that the expression of the *RrNUDX1* gene are closely related the biosynthesis of monoterpene aromatic components in *R. rugosa*.

The faster growth morphology of overexpressed petunias couldn't be explained by fragrance synthetic function of *RrNUDX1*. *NUDX1* belongs to the Nudix hydrolase gene family found in bacteria, viruses, and eukaryotes (*Schmittgen & Livak, 2008*). NUDX1 protein of *A. thaliana* can effectively remove nucleotide damage due to cells ROS oxidation which increase tolerance to adverse environments (*Ishibashi et al., 2005*; *Yoshimura & Shigeoka, 2015*) by hydrolyzing 8-oxo-(d) GTP, dNTP, NADH, and dihydroneopterin (*Ishibashi, Hayakawa & Sekiguchi, 2003*; *Klaus et al., 2005*; *Takagi et al., 2012*). The transgenetic petunias with stronger growth vigor which is associated with the improved environment tolerance be owing to nucleotide damage repair ability of overexpressed *RrNUDX1*. Anyway, 68% homology of RrNUDX1 with AtNUDX1 was just an clue of hydrolase function which need be confirmed by a follow-up study.

The petunia MD is an excellent material for verifying the function of floral genes (*Verdonk et al., 2003*), its aroma components are mainly composed of benzodiazepine components (*Schuurink, Haring & Clark, 2006*). Lvcker (*Lücker et al., 2001*) introduced the exogenous LIS gene into a non-scented petunia and found no direct effect on the production of the volatile linalool, and liquid chromatogram test showed the presence of linalool actually in the form of glycosides. In the present study, overexpression of the *RrNUDX1* gene in petunia MD did not influence the type of volatile fragrance components

of petunia. The reason for this result is probably that the *RrNUDX1* gene is mainly located in the MVA and MEP pathways, therefore, the petunia Benzene/phenylpropanoid metabolites cannot be directly used as a substrates (*Lücker et al., 2001*). However, overexpression of *RrNUDX1* gene effectively improved the accumulation of major aromatic components such as methyl benzoate in petunia flowers. These findings indicate that the *RrNUDX1* gene from traditional Chinese *R. rugosa* has the function of enhancing the fragrance of petunia flowers. And these results lay an important foundation for the homeotic transformation of *RrNUDX1* in *R. rugosa* for cultivating new rose varieties with high essential oil content.

## CONCLUSIONS

We identified the gene *RrNUDX1* from R. rugosa 'Tang Hong' whose expression pattern was correlated with the main aroma component content. The heterologously overexpressed *Petunia hybrida* have stronger growth vigor with wider blades and stronger fragrance with significantly increasing methyl benzoate. These findings indicate that the *RrNUDX1* gene play a role in enhancing the fragrance of petunia flowers. Further study of regulation of *RrNUDX1* would provide excellent genetic resources and technical means for the aroma quality improvement of other flowers.

### Funding
This research was funded by the National Key R&D Program of China, grant number 2018YFD1000400, the National Natural Science Foundation of China, grant number 31972454, 32002076 and 31772340, the Natural Science Foundation of Jiangsu province, grant number BK20200925 and the Research Project of Natural Science of Higher Education in Jiangsu province, grant number 20KJB210004. The funders had no role in study design, data collection and analysis, decision to publish, or preparation of the manuscript.

### Grant Disclosures
The following grant information was disclosed by the authors:
National Key R&D Program of China: 2018YFD1000400.
National Natural Science Foundation of China: 31972454, 32002076, 31772340.
Natural Science Foundation of Jiangsu province:  BK20200925.
Research Project of Natural Science of Higher Education in Jiangsu province: 20KJB210004.

### Competing Interests
The authors declare there are no competing interests.

### Author Contributions
- Lixia Sheng conceived and designed the experiments, prepared figures and/or tables, and approved the final draft.
- Shu Zang conceived and designed the experiments, performed the experiments, analyzed the data, prepared figures and/or tables, authored or reviewed drafts of the paper, and approved the final draft.

- Jianwen Wang analyzed the data, authored or reviewed drafts of the paper, and approved the final draft.
- Tiantian Wei performed the experiments, prepared figures and/or tables, and approved the final draft.
- Yong Xu conceived and designed the experiments, prepared figures and/or tables, authored or reviewed drafts of the paper, and approved the final draft.
- Liguo Feng conceived and designed the experiments, authored or reviewed drafts of the paper, and approved the final draft.

## DNA Deposition

The following information was supplied regarding the deposition of DNA sequences:

Sequencing data are available at GenBank: KX096710.1.

## Data Availability

Raw data is available in the Supplementary Files.

## Supplemental Information

Supplemental information for this article can be found online at http://dx.doi.org/10.7717/peerj.11098#supplemental-information.

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
