# Peer review of "Overexpression of a Rosa rugosa Thunb. NUDX gene enhances biosynthesis of scent volatiles in petunia"

_PeerJ, doi:10.7717/peerj.11098_

## Round 0.1 · original submission · Minor Revisions

The three reviewers and I find your results interesting and worthy also from a biotechnological perspective. However, for your manuscript to be accepted for publication you need to tackle convincingly all the comments and suggestions from the reviewers. Particularly relevant are those concerning the description of the experimental design, and the statistical method utilized in your work.

Reviewer 1 ·

Basic reporting

No comments

Experimental design

No comments

Validity of the findings

No comments

Additional comments

The article is devoted to the study of petunia plants transformed with the NUDX gene from rosa plants.
The authors believe that this gene is capable of increasing the content of aromatic oils and is promising for biotechnology.
The authors have done a great work, obtained important data of interest for modern plant physiology.
The experiment was well thought out, the set goals were achieved, and adequate methods were applied.
However, there are minor comments on the work.

One of the main remarks is the lack of a description of the statistical methods used by the authors in the article.

In the materials and methods, it is necessary to write in more detail how and under what conditions calli were formed, how the regeneration was carried out, to describe in more detail the conditions of electroporation and what device was used for this.

I'm not a native speaker and it's hard for me to judge, but there are a few mistakes in the text.

Reviewer 2 ·

Basic reporting

The basic reporting of the manuscript entitled 'Overexpresstion of a Rosa rugosa NUDX gene enhances biosynthesis of scent volatiles in petunia' is sound and concise. The manuscript reads well and has a great flow. The authors covered a wide range of literature in the intro. However, there are a few grammatical errors that have been highlighted in the attached document.

Experimental design

The experimental design is non existent. I have highlighted the suggestions on how to include the experimental design in the manuscript. Check the attached review document for more explanation.

Validity of the findings

This study covered a wide range of aspects from gene expression, to metabolomics, physiological and the extend of the results obtained are valid and coherent. The biological pathways investigated corresponded with the results obtained.

Additional comments

In general, the manuscript is well written and contains sound results. However, the discussion section lacks depth. I suggest that the authors validate their findings by explaining the biological processes obtained with some literature. In addition, the authors also need to compare and contrast their findings with previous studies. Further suggestions are included in the attached document.

Annotated reviews are not available for download in order to protect the identity of reviewers who chose to remain anonymous.

Reviewer 3 ·

Basic reporting

In this manuscript author's have nicely compiled their hypothesis and results to show that NUDX1 gene can be utilized for enhancing the scent of flowers in different flower species.

English need some improvement otherwise literature is cited in sufficient detail. Rest of the structure of the manuscript is fine.

Experimental design

Experimental design is well planned. Research question has approached the practical aspect of increasing demand of aromas.

Validity of the findings

The functions of the gene NUDX1 is validated in petunia to confirm the hypothesis. Conclusion and inferences made in the discussion are well supported by the results.

Additional comments

1. From the introduction what I understood that NUDX are multiple genes then what is special about targeting the NUDX1 gene? Is this based on homology with Arabidopsis or based on the information available in Magnard et al., 2015? Explaining it in introduction is important because whole story of the paper is around NUDX1.

2. Line 30: Consider using “expensive” in place of extremely expensive.

3. Line 37: Revise the sentence “Terpenes are the most abundant volatiles in floral aroma”, including…..

4. Line 153: Add citation for the software or from where you get this software.

5. Fig 2: For P2 tissue the expression is low or comparable with other remaining low expression tissue P3 but the content of citonellol is quite high. For this tissue expression level may not be consistent with content.

6. Applying statistical significance test on the expression and metabolite content is important.

---

## Round 0.2 · accepted · Accept

I checked that you complied with all the comments of the three reviewers. Your manuscript is, therefore, suitable for publication.